# Utility of adding Radiomics to clinical features in predicting the outcomes of radiotherapy for head and neck cancer using machine learning

Tarun Gangil[1], Krishna Sharan[1], B. Dinesh Rao[2‡], Krishnamoorthy Palanisamy[3], Biswaroop Chakrabarti[3‡], Rajagopal Kadavigere[4]*

1 Department of Radiotherapy and Oncology, Kasturba Medical College-Manipal, Manipal Academy of Higher Education, Manipal, Karnataka, India, 2 Manipal School of Information Sciences, Manipal Academy of Higher Education, Manipal, Karnataka, India, 3 Philips Research India, Bangalore, Karnataka, India, 4 Department of Radiology, Kasturba Medical College-Manipal, Manipal Academy of Higher Education, Manipal, Karnataka, India

☯ These authors contributed equally to this work.
‡ BDR and BC also contributed equally to this work.
* rajarad@gmail.com

**Data Availability Statement:** The data could not be shared because it is an Intellectual property of Manipal Academy of Higher Education, Manipal,

## Abstract

### Background

Radiomics involves the extraction of quantitative information from annotated Computed-Tomography (CT) images, and has been used to predict outcomes in Head and Neck Squamous Cell Carcinoma (HNSCC). Subjecting combined Radiomics and Clinical features to Machine Learning (ML) could offer better predictions of clinical outcomes. This study is a comparative performance analysis of ML models with Clinical, Radiomics, and Clinico-Radiomic datasets for predicting four outcomes of HNSCC treated with Curative Radiation Therapy (RT): Distant Metastases, Locoregional Recurrence, New Primary, and Residual Disease.

### Methodology

The study used retrospective data of 311 HNSCC patients treated with radiotherapy between 2013–2018 at our centre. Binary prediction models were developed for the four outcomes with Clinical-only, Clinico-Radiomic, and Radiomics-only datasets, using three different ML classification algorithms namely, Random Forest (RF), Kernel Support Vector Machine (KSVM), and XGBoost. The best-performing ML algorithms of the three dataset groups was then compared.

### Results

The Clinico-Radiomic dataset using KSVM classifier provided the best prediction. Predicted mean testing accuracy for Distant Metastases, Locoregional Recurrence, New Primary, and Residual Disease was 97%, 72%, 99%, and 96%, respectively. The mean area under the receiver operating curve (AUC) was calculated and displayed for all the models using three dataset groups.

India and Philips Healthcare, Bangalore. They may have future commercial interest using the dataset. The details of the person who can be contacted regarding the same is as follows: Name: Mr. Manjunatha Maiya Designation: Senior Project Manager, Philips, Bangalore email: manjunatha. maiya@philips.com. The authors had no special access privileges to the data others would not have.

**Funding:** The study was funded by Manipal Academy of Higher Education and Philips, India. The role of the funding agency was in designing the study.

**Competing interests:** The authors have declared that no competing interests exist.

**Abbreviations:** HNSCC, Squamous Cell Head and Neck Cancer; RT, Radiation Therapy; RF, Random Fores; KSVM, Kernel Support Vector Machine; AUC, Area under the Receiver operating curve; SFFS, Sequential Forward Floating selection; ML, Machine Learning; ROC, Receiver operating curve; CNN, Convolution Neural Network; CT, Computed Tomography; AI, Artificial Intelligence; PET, Positron Emission Tomography; MRI, Magnetic Resonance Imaging.

## Conclusion

Clinico-Radiomic dataset improved the predictive ability of ML models over clinical features alone, while models built using Radiomics performed poorly. Radiomics data could therefore effectively supplement clinical data in predicting outcomes.

## 1. Introduction

Head and Neck Squamous Cell Carcinoma (HNSCC) refers to a constellation of cancers in the Head and Neck region and is an important contributor to cancer-related morbidity and mortality. HNSCC is a significant health issue in India with an annual incidence of 77,000 new cases contributing to one-third of all cancers diagnosed. Treatment outcomes of HNSCC can be pretty variable, and there is scope for improvement in predicting treatment outcomes to improve survival, reduce toxicity, and further the understanding of HNSCC.

Development in the field of Artificial Intelligence (AI) has enabled the analysis of large volume data, and Machine Learning (ML) tools are being increasingly utilized in diverse fields within medicine. They are used for studying HNSCC [1]. Radiomics is an emerging method used to obtain additional information from imaging. Radiomics involves the conversion of medical images into quantitative data, thereby potentially enabling healthcare personnel in better diagnostic, therapeutic and prognostic decision-making [2]. Statistical, shape-based, histogram, and texture-based features are extracted from the Regions of Interest (ROI) for analysis and clinical correlation [3, 4]. Studies have shown that Radiomics facilitates diagnosis, treatment planning, and predicting outcomes. The development of AI, principally ML and deep learning algorithms has boosted the potential of the typically large-volume quantitative Radiomics data [5].

This research evaluates the additional benefit of Radiomics from annotated diagnostic CT images over clinical data in predicting the outcomes of HNSCC patients treated with radiotherapy. This study is an extension of our previous study [6]. It presents the change in the performance of individual models to predict Distant and Locoregional Recurrence, New Primary and Residual Disease after adding Radiomics data to the previously built models using only clinical data.

## 2. Methodology

This study developed and compared the analytics model to predict the clinical outcomes of HNSCC. The clinical information was retrieved from the hospital medical records, and Radiomics information was obtained from diagnostic contrast CT images. The study included HNSCC patients treated with curative intent radiotherapy at our centre. Though the minimum sample size was calculated to be 256 (S1B in S1 Appendix), we included all eligible patients treated between 2013–2018 [7]. Eligible cases included those treated with curative intent radiotherapy (either as definitive or as an adjuvant to surgery), completed their prescribed treatment, and had a minimum follow-up of three months post-treatment in addition to having the CT images necessary for the extraction of Radiomics. Clearance was obtained from the Institutional Ethics Committee before collecting the data from patient records, and the study was registered with the Clinical Trials Registry of India (CTRI Number: CTRI/2018/04/013517). The details of the dataset are presented in Fig 1.

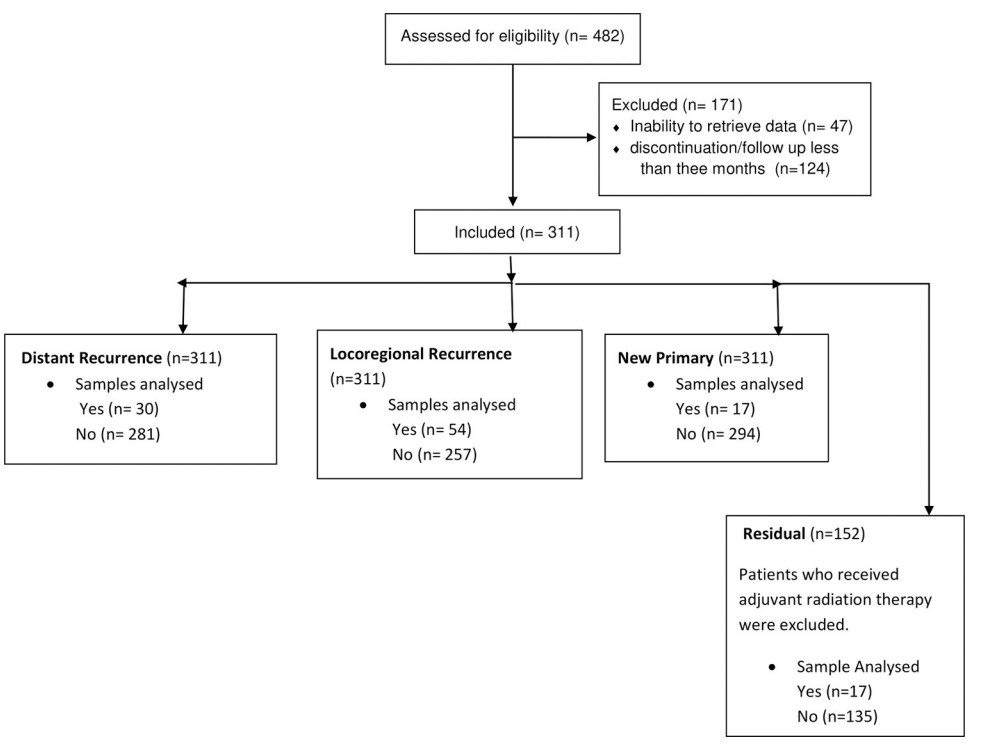

**Fig 1. Flowchart of the study.**

## 2.1. CT image acquisition

All CT scans were acquired on Incisive CT 128-slice, or Big bore Philips CT-16 slice (Philips Healthcare, Netherlands) multidetector CT scanner. The standard protocol was followed for image acquisition. The scanning parameters were namely, region- skull base down to the thoracic inlet; energy, 120 kV; milliampere-seconds, 250 mAs; pitch, 0.993; detector collimation, 0.625 mm; rotation time, 0.75s; matrix, 512×512; section thickness, 3 mm; and field-of-view, 250 mm. Only contrast-enhanced CT scans were taken for Radiomics analysis. Non-ionic iodinated contrast medium was used (Ultravist 300, Bayer Schering Pharma, Berlin, Germany, 50ml @ 4 mL/sec). Scans were obtained 70 seconds after intravenous contrast administration.

## 2.2. Radiomics data extraction

The radiation oncologist annotated the diagnostic CT images separately for the primary tumour region and lymph node/s. Radiomics features were thereafter extracted from these annotated regions. An example of annotation of tumour region and lymph node/s using open source 3D Slicer software, version 4.10.0, is shown in Fig 2.

Pyradiomics toolbox [8], provided as an extension in the 3D-Slicer tool, was used to extract Radiomics features from the gross tumour and lymph nodes. The extracted features were structured in columns as inputs for each model. The data constituted of quantitative features from medical images using various data characterization algorithms, including quantitative information on Region of Interest (ROI) for statistical-based, transform-based, model-based, and shape-based features [3]. The Radiomics dataset included:

- Shape-based features: 14

- Gray-level Dependence matrix features: 14

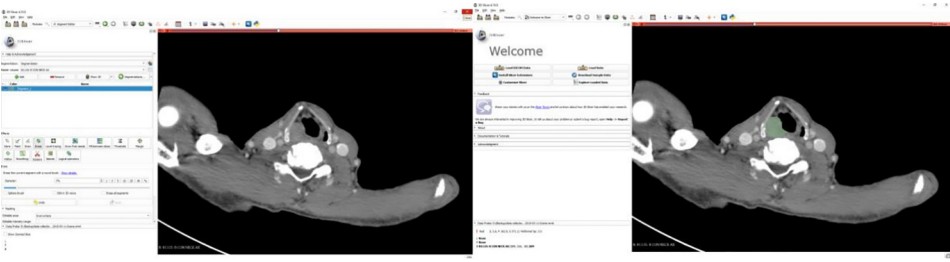

CT Image slice                                 Annotated Image for tumour region

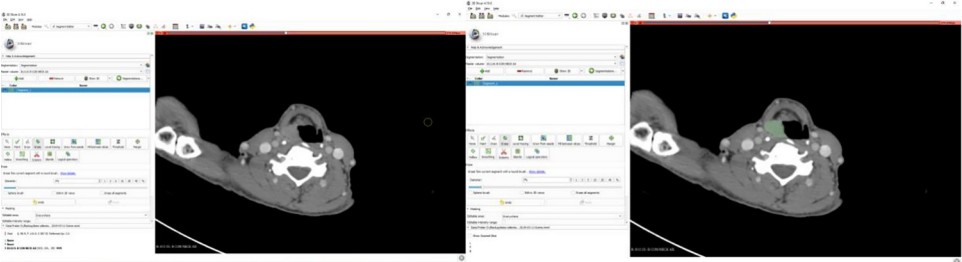

CT Image slice                                 Annotated Image for tumour region

a)

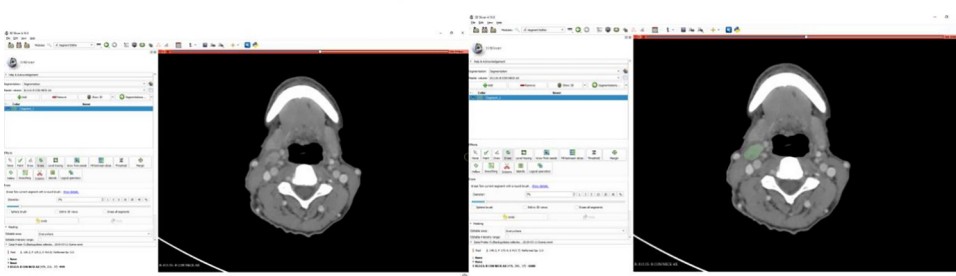

CT Image slice                          Annotated Image for lymph node region

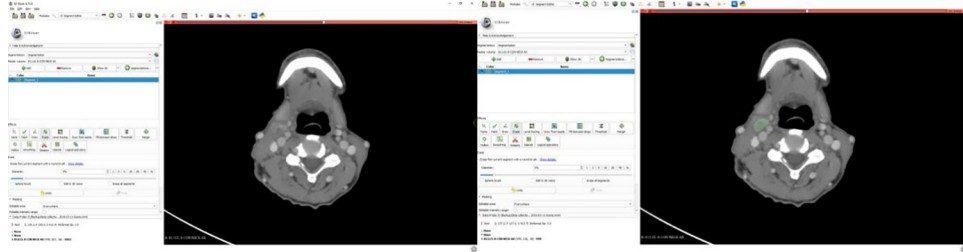

CT Image slice                          Annotated Image for lymph node region

b)

**Fig 2.** Illustration of original contrast CT-slices in axial view and its respective annotated slice using 3D slicer; a) Tumour region b) Lymph nodes region.

- Gray-level Cooccurence matrix features: 24

- First-order statistics features:18

- Gray-level run length matrix features:16

- Gray-level size zone matrix: 16

- Neighbouring gray-tone difference matrix features: 5

## 2.3. Workflow of the study

**2.3.1 Structuring of the data.**   The clinical factors and Radiomics data were structured in separate non-mutually exclusive columns. The Radiomics dataset was placed alongside the clinical features [6] in separate columns, making a total of 602 columns (388 Clinical and 214 Radiomics features) and 311 rows (samples).

**2.3.2 Data pre-processing.**   The raw collected data must be pre-processed to make it suitable for ML algorithms. The collected data consisted of missing values, necessitating imputed corrections beforehand. We used iterative imputation techniques [6] to impute the missing values. To handle class-imbalance in the dataset, minority oversampling techniques such as Randomoversampler, Synthetic Minority Oversampling Technique (SMOTE), BorderlineS-MOTE, Support Vector Machine SMOTE (SVM SMOTE), and Adaptive Synthetic (ADA-SYN) were used [6]. To build each model, all five minority oversampling techniques were used; the results from the best-performing technique are reported.

After performing minority oversampling on the original dataset, the synthetic samples were separated from the original samples. The original samples were divided into two parts: majority and minority, based on class-labels. In our case, patients without any recurrence constituted the majority (class-label 0). The training-testing split was then performed on samples from majority-class using 70:30 ratio. To build the ML models, the training dataset was assembled in such a way that it would contain train-split and synthetic samples, and testing dataset would unconditionally include all original minority samples along with test-split.

Thereafter, the training dataset was scaled to zero mean and unit standard deviation using standard scalar function. Due to curse-of-dimensionality resulting from a large number of variables in relation to sample size, feature selection algorithms, namely Sequential Forward Floating selection (SFFS) and Boruta algorithms, were implemented on the dataset. The best k-features based on the highest accuracy (hyperparameter of SFFS) scores were chosen to build ML models.

**2.3.3 Fitting model on the training dataset.**   Predictive ML models using Radiomics alone and combined Clinical-Radiomics data were then built to predict four clinical outcomes: Distant Recurrence, Locoregional Recurrence, Residual Disease, and development of New Primary. RF, KSVM and XGBoost algorithms were used for fitting onto the training dataset. The selected features were made to fit on the dataset, and the mean training accuracy was calculated for ten iterations.

**2.3.4 Evaluation using test-split.**   The performance of the designed classifier was evaluated using the testing dataset. Each value of the performance metrics [9] was reported as the mean of ten iterations.

**2.3.5 Visualising ROC plots and performance comparison.**   Stratified k-fold cross-validation was performed for ten folds, and its plots were visualized for each model to ensure consistency of the performance reported throughout the dataset.

The best-performing ML model for each group- Clinical-only, Radiomics-only, and combined Clinical-Radiomics datasets- was identified based on accuracy, macro and weighted F-1

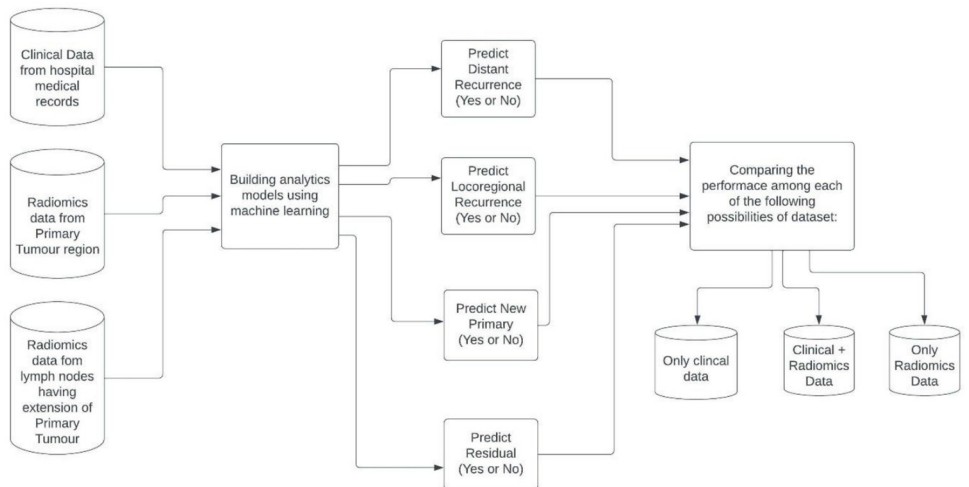

**Fig 3. Flowchart describing workflow of the study using Clinical, Clinico-Radiomic and Radiomics datasets.**

score, and area under the Receiver Operating Characteristic (ROC) curve (Fig 3). The representative ML model for each of the three dataset groups was then compared to determine if Radiomics could supplant or replace clinical data (S1C in S1 Appendix).

## 3. Results

A total of 311 patients with HNSCC were included in the study. The mean/median follow-up duration of the cohort was 18 months (3–85.1 months). The details of the dataset are summarized in Table 1. The Radiomics Quality score (RQS) was calculated to be 45% (S1A in S1 Appendix) [10]. The best-performing ML algorithm with each of the three datasets is compared and reported below.

The SFFS algorithm, and utilizing accuracy score for selecting k_features, performed better than Boruta for all three datasets in the optimal feature selection. Also, KSVM performed better than RF and XGBoost while implementing the ML algorithm on the datasets.

It was observed that the model developed using Radiomics alone performed the poorest for all four clinical outcomes. In contrast, the Clinico-Radiomics dataset provided the best predictive performance. The persistence of residual disease on completion of treatment was the most challenging clinical outcome to predict, and the Radiomics-alone dataset performed poorly in this regard.

With Clinico-Radiomic data, the mean training and testing accuracy were calculated to be 100% and 97% for Distant Metastases, 100% and 72% for Locoregional Recurrence, 100% and 99% for development of New Primary, and 100% and 96% for persistent Residual Disease. Sensitivity and specificity were calculated to be 96% and 98% for Distant Metastases, 100% and 82% for Locoregional Recurrence, 94% and 100% for New Primary, and 94% and 100% for Residual Disease, respectively. Mean ROC was calculated to be 99%, 94%, 99%, and 98% for Distant Metastases, Locoregional Recurrence, New Primary, and Residual Disease, respectively. Further, the performance of the individual class of the designed binary classifier model was measured for training and testing datasets using a weighted F1-score for imbalanced classes. It was calculated to be typically over 90%, except for Locoregional Recurrence, where the score was only 70% (Tables 2–5), despite evidence of overfitting. In order to reduce overfitting here, we attempted reiterating after changing learning-rate hyperparameters of SFSS and also of KSVM and RF, with no significant improvement.

**Table 1. Demographics of the retrospectively collected data of HNSCC (n = 311).**

| Patient Variable | Frequency (%) | Frequency of any recurrence (n = 96) | p-value |
|---|---|---|---|
| Age in years | Mean: 56.5 (Range:22–92) | | |
| < 60 years | 178 (57.2) | 59 (33.1%) | 0.32 |
| ≥ 60 years | 133(42.8) | 37(27.8%) | |
| Gender: | | | |
| Female | 54 (17.4) | 17(31.5%) | 0.51 |
| Male | 257 (82.6) | 79(30.7%) | |
| Site: | | | |
| Oral Cavity | 156 (50.2) | 44(28.2%) | 0.52 |
| Oropharynx | 38 (12.2) | 13(34.2%) | |
| Larynx, Hypopharynx | 112 (36) | 37(33%) | |
| Other sites | 5 (1.6) | 2(40%) | |
| Staging: | | | |
| T1 | 35 (11.3) | 15(42.8%) | 0.33 |
| T2 | 98 (31.6) | 26(26.5%) | |
| ≥T3 | 178 (57.1) | 55(30.8%) | |
| N0 | 120 (38.6) | 38(31.6%) | 0.28 |
| N1 | 91 (29.2) | 19(20.8%) | |
| ≥ N2 | 100 (32.2) | 39(39%) | |
| Group: | | | |
| I -II | 86 (27.6) | 20(23.2%) | 0.56 |
| ≥ III | 225 (72.4) | 76(33.7%) | |
| Radiotherapy type: | | | |
| Definitive | 152 (48.9) | 53(34.8%) | 0.47 |
| Adjuvant | 159 (51.1) | 43(27.0%) | |
| Concurrent Chemotherapy: | | | |
| Yes | 173 (55.6) | 58(33.5%) | 0.15 |
| No | 138 (44.4) | 38(27.5%) | |

**Table 2. Comparative results for Distant Recurrence (311 samples).** The column of the model utilizing only clinical data from our previous study [6] is included for comparison.

| Model Name | Distant Recurrence | | |
|---|---|---|---|
| Dataset Type | Only Clinical | Clinico-Radiomic | Only Radiomics |
| Total number of independent variables (columns) | 388 | 602 | 214 |
| Best performing ML algorithm | KSVM | KSVM | KSVM |
| Minority Oversampling method | ADASYN | SMOTE | ADASYN |
| No. of samples after oversampling | 573 | 562 | 571 |
| No. of synthetic samples | 262 | 251 | 260 |
| No. of features selected by SFFS | 18 | 42 | 23 |
| No. of original samples for class-0 (no recurrence) | 281 | 281 | 281 |
| No. of original samples for class -1 (recurrence) | 30 | 30 | 30 |
| No. of samples in Training dataset for class 0 (no recurrence) | 188 | 188 | 188 |
| No. of samples in Training dataset for class 1 (recurrence) | 262 | 251 | 260 |
| No. of samples in Testing dataset for class 0 (no recurrence) | 93 | 93 | 93 |
| No. of samples in Testing dataset for class 1 (recurrence) | 30 | 30 | 30 |
| Mean training accuracy (CI) | 0.99 (0.989–0.991) | 1 (1.000–1.000) | 0.87 (0.870–0.870) |
| Mean testing accuracy (CI) | 0.94 (0.938–0.942) | 0.97 (0.968–0.972) | 0.84 (0.829–0.851) |

*(Continued)*

**Table 2.** (Continued)

| Model Name | Distant Recurrence | | |
|---|---|---|---|
| Dataset Type | Only Clinical | Clinico-Radiomic | Only Radiomics |
| Sensitivity (CI) | 0.87 (0.869–0.871) | 0.96 (0.958–0.962) | 0.95 (0.949–0.951) |
| Specificity (CI) | 0.96 (0.959–0.961) | 0.98 (0.978–0.982) | 0.92 (0.919–0.921) |
| Training F1 score Class 0 (no recurrence) | 0.99 (0.998 – 0.998) | 1(0.999 – 0.999) | 0.84 (0.835 – 0.847) |
| Training F1 score Class 1 (recurrence) | 0.99 (0.998 – 0.998) | 1(0.999 – 0.999) | 0.89 (0.883 – 0.895) |
| Macro Training F1 score (CI) | 0.96 (0.958–0.962) | 0.98 (0.979–0.981) | 0.85 (0.849–0.851) |
| Weighted Training F1 score (CI) | 0.99(0.989 – 0.990) | 1(0.996 – 1.000) | 0.87 (0.868–0.873) |
| Testing F1 score Class 0 (no recurrence) | 0.97(0.975 – 0.977) | 0.98(0.977 – 0.989) | 0.88 (0.873–0.882) |
| Testing F1 score Class 1 (recurrence) | 0.91(0.905 – 0.917) | 0.94(0.933 – 0.945) | 0.75 (0.741–0.752) |
| Macro Testing F1 score (CI) | 0.93 (0.928–0.932) | 0.95 (0.948–0.952) | 0.80 (0.798–0.802) |
| Weighted Testing F1 score (CI) | 0.96 (0.958 – 0.969) | 0.97(0.962–0.971) | 0.85 (0.843–0.856) |
| Base Algorithm for SFFS | KSVM | KSVM | KSVM |
| Mean AUC_ROC(CI) | 0.97 (0.969–0.971) | 0.99 (0.989–0.991) | 0.79 (0.783–0.797) |

**Table 3. Comparative results for Locoregional Recurrence (311 samples).** The column of the model utilizing only clinical data from our previous study [6] is included for comparison.

| Model Name | Locoregional Recurrence | | |
|---|---|---|---|
| Dataset Type | Only Clinical | Clinico- Radiomics | Only Radiomics |
| Total number of independent variables (columns) | 384 | 598 | 214 |
| Best performing ML algorithm | KSVM | KSVM | RF |
| Minority Oversampling method | SMOTE | BorderlineSMOTE | SMOTE |
| No. of samples after oversampling | 514 | 514 | 514 |
| No. of synthetic samples | 203 | 203 | 203 |
| No. of features selected by SFFS | 24 | 89 | 16 |
| No. of original samples for class-0 (no recurrence) | 257 | 257 | 257 |
| No. of original samples for class -1 (recurrence) | 54 | 54 | 54 |
| No. of samples in Training dataset for class 0 (no recurrence) | 172 | 172 | 172 |
| No. of samples in Training dataset for class 1(recurrence) | 203 | 203 | 203 |
| No. of samples in Testing dataset for class 0 (no recurrence) | 85 | 85 | 85 |
| No. of samples in Testing dataset for class 1 (recurrence) | 54 | 54 | 54 |
| Mean training accuracy (CI) | 0.96 (0.958–0.962) | 1 (1.000–1.000) | 0.87 (0.869 – 0.871) |
| Mean testing accuracy (CI) | 0.73 (0.718–0.742) | 0.72 (0.718 – 0.722) | 0.71 (0.706 – 0.714) |
| Sensitivity (CI) | 0.83 (0.828 – 0.832) | 1 (1.000–1.000) | 0.62 (0.618 – 0.622) |
| Specificity (CI) | 0.73 (0.728–0.732) | 0.82 (0.818–0.822) | 0.83 (0.828–0.832) |
| Training F1 score Class 0 (no recurrence) | 0.97 (0.964–0.976) | 1 (0.996–1.000) | 0.83 (0.813–0.835) |
| Training F1 score Class 1 (recurrence) | 0.97 (0.952–0.975) | 1 (1.000–1.000) | 0.89 (0.872–0.892) |
| Macro Training F1 score (CI) | 0.95 (0.945–0.955) | 0.98 (0.978–0.982) | 0.85 (0.848–0.852) |
| Weighted Training F1 score (CI) | 0.97 (0.961–0.976) | 1 (1.000–1.000) | 0.86 (0.852–0.869) |
| Testing F1 score Class 0 (no recurrence) | 0.82 (0.813–0.821) | 0.63 (0.625–0.638) | 0.73 (0.724–0.740) |
| Testing F1 score Class 1(recurrence) | 0.69 (0.682–0.693) | 0.72 (0.718–0.736) | 0.68 (0.674–0.692) |
| Macro Testing F1 score (CI) | 0.69 (0.687–0.693) | 0.70 (0.697–0.703) | 0.69 (0.688–0.692) |
| Weighted Testing F1 score (CI) | 0.77 (0.752–0.779) | 0.66 (0.652–0.669) | 0.71 (0.701–0.721) |
| Base Algorithm | KSVM | KSVM | RF |
| Mean AUC_ROC(CI) | 0.73 (0.729–0.731) | 0.94 (0.938–0.942) | 0.78 (0.776–0.784) |

**Table 4. Comparative results for New Primary (311 samples).** The column of model utilizing only clinical data from our previous study [6] is included for comparison.

| Model Name | New Primary | | |
|---|---|---|---|
| Dataset Type | Only Clinical | Clinico- Radiomics | Only Radiomics |
| Total number of independent variables (columns) | 388 | 602 | 214 |
| Best performing ML algorithm | KSVM | KSVM | KSVM |
| Minority Oversampling method | SMOTE | ADASYN | ADASYN |
| No. of samples after oversampling | 588 | 594 | 587 |
| No. of synthetic samples | 277 | 283 | 276 |
| No. of features selected by SFFS | 42 | 43 | 53 |
| No. of original samples for class-0 (no recurrence) | 294 | 294 | 294 |
| No. of original samples for class -1 (recurrence) | 17 | 17 | 17 |
| No. of samples in Training dataset for class 0 (no recurrence) | 196 | 196 | 196 |
| No. of samples in Training dataset for class 1 (recurrence) | 277 | 283 | 276 |
| No. of samples in Testing dataset for class 0 (no recurrence) | 98 | 98 | 98 |
| No. of samples in Testing dataset for class 1 (recurrence) | 17 | 17 | 17 |
| Mean training accuracy (CI) | 1 (1.000–1.000) | 1 (1.000–1.000) | 0.87 (0.869–0.871) |
| Mean testing accuracy (CI) | 0.96 (0.959–0.961) | 0.99 (0.989–0.991) | 0.78 (0.776–0.784) |
| Sensitivity (CI) | 0.94 (0.938–0.942) | 0.94 (0.938–0.942) | 0.46 (0.457–0.463) |
| Specificity (CI) | 0.98 (0.978–0.982) | 1 (1.000–1.000) | 1.0 (0.999–1.000) |
| Training F1 score Class 0 (no recurrence) | 1 (1.000–1.000) | 1 (1.000–1.000) | 0.82 (0.813–0.829) |
| Training F1 score Class 1 (recurrence) | 1 (1.000–1.000) | 1 (1.000–1.000) | 0.90 (0.889–0.910) |
| Macro Training F1 score (CI) | 0.99 (0.987–0.993) | 0.99 (0.987–0.993) | 0.85 (0.848–0.852) |
| Weighted Training F1 score (CI) | 1 (1.000–1.000) | 1 (1.000–1.000) | 0.87 (0.865–0.876) |
| Testing F1 score Class 0 (no recurrence) | 0.99 (0.986–0.991) | 0.99 (0.986–0.990) | 0.85 (0.843–0.852) |
| Testing F1 score Class 1 (recurrence) | 0.95 (0.948–0.952) | 0.98 (0.978–0.982) | 0.57 (0.562–0.583) |
| Macro Testing F1 score (CI) | 0.92 (0.917–0.923) | 0.97 (0.967–0.973) | 0.70 (0.698 – 0.702) |
| Weighted Testing F1 score (CI) | 0.98 (0.978–0.982) | 0.99 (0.987–0.993) | 0.81 (0.806–0.820) |
| Base Algorithm for SFFS | KSVM | KSVM | KSVM |
| Mean AUC_ROC (CI) | 0.98 (0.978–0.982) | 0.99 (0.990–0.990) | 0.80 (0.796–0.804) |

ROC curves were obtained to visualize the results. Ten-split stratified k-fold cross-valida-tion was performed using the best combination of ML workflow identified to ensure consis-tency of reported ROC values (Fig 4).

The ROC value using only the clinical dataset was 0.99 (±0.00), 0.96 (±0.02), 0.99 (±0.00) and 0.98 (±0.03), respectively, for Distance Recurrence, Locoregional Recurrence, New Pri-mary, and Residual Disease. Similarly, using the Clinico-Radiomics dataset, the ROC values were 0.99 (±0.00), 0.98 (±0.01), 0.99 (±0.00) and 0.95 (±0.05), respectively. Using only Radio-mics, the ROC values were the lowest, with values of 0.74 (±0.10), 0.85 (±0.07), 0.82 (±0.06) and 0.73 (±0.16).

## 4. Discussion

Accurate prediction of eventual outcomes carries great importance in medicine, where out-comes can be pretty variable across patients, scenarios, and treatments. Radiomics has an unexplored potential in clinical prediction and has been utilized in various degrees in radio-therapy for HNSCC, including tumour segmentation, prognostication, and monitoring of changes in normal tissues following radiotherapy [11]. Recent developments in AI have expo-nentially enhanced the ability to analyze large volume data, further increasing the potential of Radiomics. However, it has already been reasoned that it would be best to use Radiomics along

**Table 5. Comparative results for Residual Disease (152 samples).** The column of the model utilizing only clinical data from our previous study [6] is included for comparison.

| Model Name | Residual Disease | | |
| --- | --- | --- | --- |
| Dataset Type | Only Clinical | Clinico- Radiomics | Only Radiomics |
| Total number of independent variables (columns) | 354 | 568 | 214 |
| Best performing ML algorithm | KSVM | KSVM | KSVM |
| Minority Oversampling method | SMOTE | ADASYN | ADASYN |
| No. of samples after oversampling | 270 | 270 | 270 |
| No. of synthetic samples | 118 | 118 | 118 |
| No. of features selected by SFFS | 312 | 98 | 15 |
| No. of original samples for class-0 (no recurrence) | 135 | 135 | 135 |
| No. of original samples for class -1 (recurrence) | 17 | 17 | 17 |
| No. of samples in Training dataset for class 0 (no recurrence) | 90 | 90 | 90 |
| No. of samples in Training dataset for class 1 (recurrence) | 118 | 118 | 118 |
| No. of samples in Testing dataset for class 0 (no recurrence) | 45 | 45 | 45 |
| No. of samples in Testing dataset for class 1(recurrence) | 17 | 17 | 17 |
| Mean training accuracy (CI) | 1.0 (0.999–1.001) | 1.0 (0.999–1.001) | 0.88 (0.879–0.881) |
| Mean testing accuracy (CI) | 0.92 (0.906–0.914) | 0.96 (0.957–0.963) | 0.83 (0.822–0.838) |
| Sensitivity (CI) | 0.89 (0.886–0.894) | 0.94 (0.937–0.943) | 0.85 (0.848–0.852) |
| Specificity (CI) | 1.0 (0.999–1.000) | 1.0 (0.999–1.000) | 0.88 (0.877–0.883) |
| Training F1 score Class 0 (no recurrence) | 1 (0.999–1.000) | 1 (0.999–1.000) | 0.86 (0.852–0.862) |
| Training F1 score Class 1 (recurrence) | 1 (0.999–1.000) | 1 (0.999–1.000) | 0.9 (0.899–0.912) |
| Macro Training F1 score (CI) | 0.99 (0.986–0.994) | 0.99 (0.986–0.994) | 0.85 (0.847–0.853) |
| Weighted Training F1 score (CI) | 1 (0.999–1.000) | 1 (0.999–1.000) | 0.88 (0.874–0.882) |
| Testing F1 score Class 0 (no recurrence) | 0.95 (0.939–0.954) | 0.98 (0.965–0.986) | 0.87 (0.862–0.878) |
| Testing F1 score Class 1 (recurrence) | 0.88 (0.876–0.892) | 0.94 (0.934–0.954) | 0.75 (0.749–0.763) |
| Macro Testing F1 score (CI) | 0.89 (0.886–0.894) | 0.95 (0.944–0.956) | 0.80 (0.797–0.803) |
| Weighted Testing F1 score (CI) | 0.93 (0.924–0.931) | 0.97 (0.968–0.985) | 0.84 (0.832–0.841) |
| Base Algorithm for SFFS | KSVM | KSVM | KSVM |
| Mean AUC_ROC(CI) | 0.99 (0.986–0.994) | 0.98 (0.977– 0.983) | 0.85 (0.838–0.862) |

with additional information, including clinical features, rather than in isolation to maximize the potential of predictive modeling [12]. Therefore, this study was conducted assuming that Radiomics when added to Clinico-Pathological data would improve the performance of ML models in predicting outcomes of HNSCC treated with RT.

Clinico-Radiomic dataset performed the best in all the four outcome predictions referred to in this study. Despite the models working on clinical data alone are already providing excellent predictive abilities in our study, adding Radiomics further improved the predictive ability.

Radiomics have been studied for a myriad of applications across all disciplines involved in HNSCC management, including molecular characterization [13], image segmentation [14], pre-surgical decision making [15], prognostication, improving treatment quality and efficiency, etc. However, most prognostication studies on Radiomics in HNSCC have utilized PET-CT scans and MRIs to a lesser extent, given the wealth of biological and chemical data captured in these images. In contrast, Radiomics has been used on diagnostic CT scans because these are the most widely available scans used to evaluate the extent and planning of treatment. CT scan Radiomics have been studied to a lesser extent with conflicting conclusions. For example, Ger et al., concluded that neither CT nor PET was independently capable of predicting survival in HNSCC [16], while Cozzi et al., were able to successfully predict

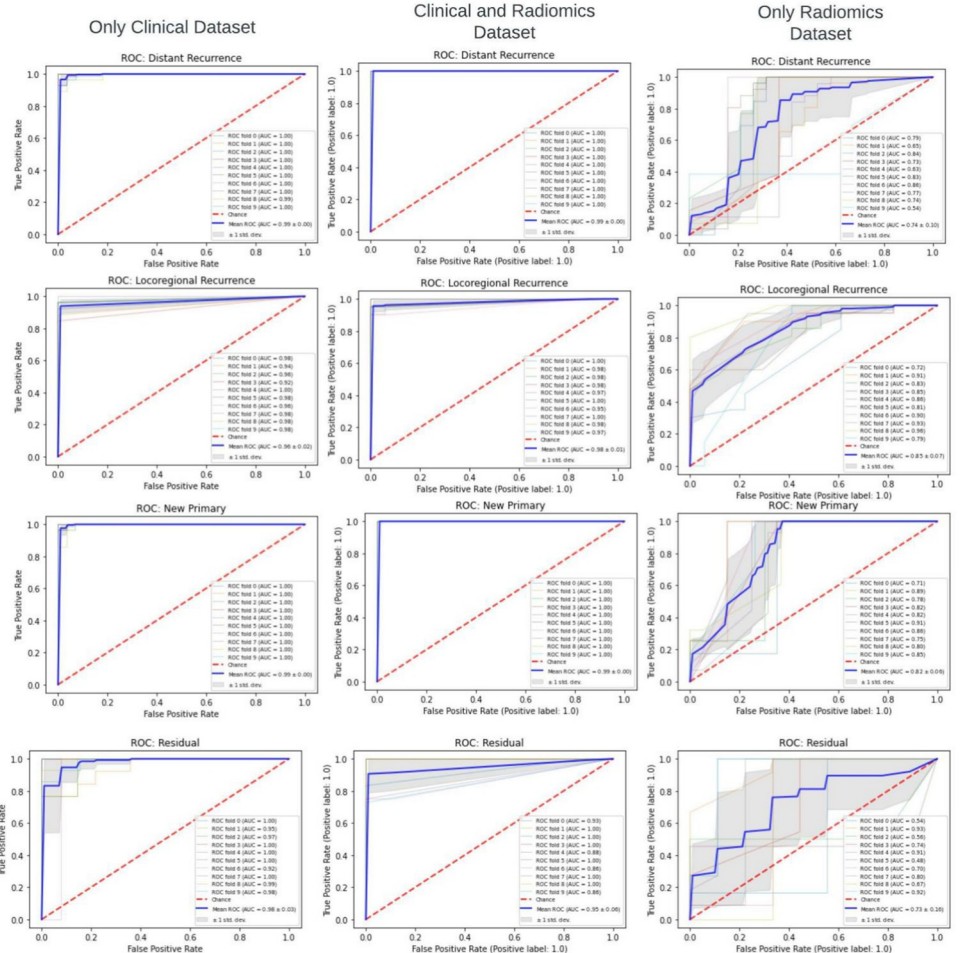

**Fig 4. ROC curves comparing the ML performance with three datasets groups (the thin line represents each iteration's ROC, and the thick line is the mean ROC).** The column of the model utilizing only clinical data from our previous study [6] is included for comparison.

survival and local control in a retrospective series of 110 patients with HNSCC treated with radiotherapy [17].

However, fewer studies have attempted to combine Radiomics with AI to enhance its potential. For example, a survey by Diamant et al., was able to successfully predict distant metastases in HNSCC by applying CNN to Radiomics data [18]. A systematic review by Giraud et al., summarizes some possible applications in HNSCC [19]. More recent reviews have looked explicitly at this integrated application for predicting disease outcomes and treatment toxicities [20, 21]. Despite promising reports, they all agree on the need for prospective validation studies prior to clinical translation.

The ML algorithms gave the best performance when fitted onto the dataset under optimal hypermeter settings that were obtained using grid search [22]. Also, the feature selection method SFFS [23] automatically selected k_best features from an exhaustive list of variables so that features with the best accuracy scores are chosen for model-building, thereby preventing the curse of dimensionality [24].

While it provides proof-of-concept of integrated Clinical-Radiomic predictive ML models, this study has several limitations. For one, it is a single-center study that relied on retrospective

data with small sample size and a considerably limited follow-up duration. Second, it used only contrast CT images, which can add a degree of heterogeneity to the collected Radiomics data. Radiomics naturally demands high-quality, standardized imaging for optimum performance [25]. Factors such as motion artifacts, time-lapse between contrast administration and image acquisition, image processing algorithms, etc., have been shown to affect data. The study also does not work to determining the mechanisms underlying the possible correlation between radiological findings and outcomes. Moreover, as with most other research [26] on Radiomics and AI, the findings need multicentric protocol-constrained imaging and prospective validation prior to routine use. Developing healthcare data management systems with the ability to recognize, classify and segregate data in real-time could greatly expedite the implementation of such AI applications in the clinic.

## 5. Conclusion

Radiomics features supplemented with clinical features predicted the clinical outcomes of HNSCC treated with radiotherapy to a high degree of accuracy. In contrast, using only Radiomics data as input offered suboptimal performance. For all the models, KSVM performed better than either RF or XGBoost. Weighted average training and testing F1-scores were equally good with clinical and Clinico-Radiomic datasets but were poor with only the Radiomics dataset. We conclude that only Radiomics data could be insufficient in predicting the HNSCC outcomes, while Clinico-Radiomic data can provide predictions of clinical utility. With prospective validation, such predictive models can be of great utility for clinical exploitation.

## Supporting information

**S1 Checklist. TREND statement checklist.**
(PDF)

**S1 Appendix.**
(PDF)

**S1 Protocol.**
(PDF)

## Acknowledgments

The authors would like to acknowledge Mr. Manjunatha Maiya, Project Manager, Philips Research India, Bangalore, Karnataka, Mr. Prasad R V, Program Manager Philips Research India, Bangalore, Karnataka, and Mr. Shrinidhi G. C., Chief Physicist, Dept. of Radiotherapy, KMC, Manipal for their help and support in the conduct of this research.

## Author Contributions

**Conceptualization:** Tarun Gangil, Krishna Sharan.

**Data curation:** Tarun Gangil, Krishna Sharan, Rajagopal Kadavigere.

**Formal analysis:** Tarun Gangil.

**Funding acquisition:** Biswaroop Chakrabarti.

**Investigation:** Tarun Gangil, Rajagopal Kadavigere.

**Methodology:** Tarun Gangil, Krishna Sharan, B. Dinesh Rao, Krishnamoorthy Palanisamy, Rajagopal Kadavigere.

**Project administration:** Krishna Sharan, Rajagopal Kadavigere.

**Resources:** Krishna Sharan, B. Dinesh Rao.

**Software:** Tarun Gangil, Krishnamoorthy Palanisamy.

**Supervision:** Krishna Sharan, B. Dinesh Rao, Krishnamoorthy Palanisamy, Rajagopal
Kadavigere.

**Validation:** Tarun Gangil.

**Visualization:** Tarun Gangil.

**Writing – original draft:** Tarun Gangil, Krishna Sharan.

**Writing – review & editing:** Krishna Sharan, B. Dinesh Rao, Krishnamoorthy Palanisamy,
Biswaroop Chakrabarti, Rajagopal Kadavigere.

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
