## [Decision Letter · Decision Letter 0]

20 Jul 2022

PONE-D-22-16089Utility of adding Radiomics to clinical features in predicting outcomes of radiotherapy for Head and Neck Cancer using Machine Learning

PLOS ONE

Dear Dr. Kadavigere,

Thank you for submitting your manuscript to PLOS ONE. After careful consideration, we feel that it has merit but does not fully meet PLOS ONE’s publication criteria as it currently stands. Therefore, we invite you to submit a revised version of the manuscript that addresses the points raised during the review process.

Dear authors,

the reviewers have now completed their reports and found that your manuscript in its present for is not suitbale for publication.

There are some major concerns that have to be answered if you want to resubmit it:

1) There are some ethical concerns, as you state that no consent was acquired from the participants for use of the data because "the study was retrospective"

2) The methodology used is not described sufficiently

3) There is no split in training and validation set and maybe an additional cohort should be analyzed

4) The english language used is unacceptable and the manuscript should be edited by a native speaker/ scientific writer

We look forward to receiving your revised manuscript.

Kind regards,

Panagiotis Balermpas

Academic Editor

PLOS ONE

Journal Requirements:

A clean copy of the edited manuscript (uploaded as the new *manuscript* file).

4. Please provide additional details regarding participant consent. In the Methods section, please ensure that you have specified (1) whether consent was informed and (2) what type you obtained (for instance, written or verbal). If your study included minors, state whether you obtained consent from parents or guardians. If the need for consent was waived by the ethics committee, please include this information.

Reviewers' comments:

Reviewer's Responses to Questions

**Comments to the Author**

1. Is the manuscript technically sound, and do the data support the conclusions?

Reviewer #1: Partly

Reviewer #2: Yes

2. Has the statistical analysis been performed appropriately and rigorously? 

Reviewer #1: N/A

Reviewer #2: Yes

3. Have the authors made all data underlying the findings in their manuscript fully available?

Reviewer #1: No

Reviewer #2: No

4. Is the manuscript presented in an intelligible fashion and written in standard English?

Reviewer #1: No

Reviewer #2: Yes

5. Review Comments to the Author

Reviewer #1: Dear authors,

1. I suggest that a native speaker proofreads the manuscript. There are several grammar mistakes, and the general style could be improved.

2. Please check the row for age in the patient characteristics table.

3. Even though you point to a previous study, I find the paper is lacking some high-level explanation of the methodology (how was the train/test split performed, which feature selection algorithms were employed, what are the weights in the F1-score computation, etc.). Right now it is very hard for the reader to understand what methodology was used in this study.

4. I have concerns whether the follow-up time is too little to be clinically significant.

Reviewer #2: Thanks for the interesting article evaluating the value of radiomics in CT imaging.

I have a couple of questions mainly on the methodolgy, which need to be clarified.

1) How was the annotation of the PT and the LN done? Manually? By whom? was each LN delineated separatly?

2)PLease provide more details on the extraction of features:

- how many features were extracted per feature type?

-Which pre-processing was done?

- Were cubic voxels used?

- Which binning method?

3) Why did you not split the dataset into a trianing and validation dataset? Since this is not performed a cross-validation is needed.

To better adress all these questions please determine the radiomics quality score:

https://www.radiomics.world/rqs

and put your score to the manuscript including your answers as an appendix

Results:

With clinico-radiomic data, the mean training and testing accuracy -> From Materials and Methods it is not clear that you have splitted in training and test set, please explain.

All your results on accuracy, sensitivity,... need confidence intervals.

6. PLOS authors have the option to publish the peer review history of their article (what does this mean?). If published, this will include your full peer review and any attached files.

Reviewer #1: No

Reviewer #2: No

---

## [Author Response · Author response to Decision Letter 0]

2 Sep 2022

Date: 16th August 2022 

Subject: Response to the reviewers’ queries and suggestions [PONE-D-22-16089]

(PONE-D-22-16089: Utility of adding Radiomics to clinical features in predicting outcomes of radiotherapy for Head and Neck Cancer using Machine Learning)

The authors would like to express their profuse gratitude for the reviewers’ time and efforts, and for the excellent criticisms raised in order to facilitate the betterment of our article. The queries have been addressed in the manuscript, and the replies are stated below:

Queries summarized (editor):

1) There are some ethical concerns, as you state that no consent was acquired from the participants for use of the data because "the study was retrospective"

Ethical clearance for this study was obtained from our Institutional Ethics Committee (approval no. 165/2018). Since the study involved retrospective data retrieved from medical records/imaging archives, and the patients were not contacted, participation consent couldn’t be taken, and was waived off by the IEC.

2) The methodology used is not described sufficiently

The methodology section has been updated to provide greater details in the original manuscript. (section 2.3.1-2.3.6)(page-3-6)

3) There is no split in training and validation set and maybe an additional cohort should be analysed.

Multiple iterations of Training and testing splits at 70:30 ratio was performed within the collected dataset. The details have been added and clarified in the revised manuscript (section 2.3.3-2.3.5)(page 5-6)

4) The english language used is unacceptable and the manuscript should be edited by a native speaker/ scientific writer.

The manuscript has been extensively re-written to address the grammatical and clarity errors, and has been screened through language-analysis software (professional version of Grammarly) for inaccuracies. 

Other observations: Your ethics statement should only appear in the Methods section of your manuscript. If your ethics statement is written in any section besides the Methods, please move it to the Methods section and delete it from any other section. Please ensure that your ethics statement is included in your manuscript, as the ethics statement entered into the online submission form will not be published alongside your manuscript. 

Statement moved to the methodology section as suggested.(page 4-5)

Other observations: Please provide additional details regarding participant consent. In the Methods section, please ensure that you have specified (1) whether consent was informed and (2) what type you obtained (for instance, written or verbal). If your study included minors, state whether you obtained consent from parents or guardians. If the need for consent was waived by the ethics committee, please include this information.

The data being retrospective, informed consent from the patients was waived off by the Institutional Ethics Committee. Its details are mentioned in the methodology section of manuscript. (page 4-5)

Other observations: We note that you have indicated that data from this study are available upon request. PLOS only allows data to be available upon request if there are legal or ethical restrictions on sharing data publicly. For information on unacceptable data access restrictions, please see http://journals.plos.org/plosone/s/data-availability#loc-unacceptable-data-access-restrictions. 

The collected data is the intellectual property of Manipal Academy of Higher Education, Manipal (India), and Philips, Bangalore (India), and as per the exhibit, we are not permitted to share the collected data. Moreover, we don’t have the government regulatory body (Health Ministry Screening Committee, Indian Council of Medical Research) approval for data sharing.

Reviewers' comments to the authors:

Reviewer #1: 

1. I suggest that a native speaker proofreads the manuscript. There are several grammar mistakes, and the general style could be improved.

The manuscript has been extensively re-written to address the grammatical and clarity errors, and has been screened through language-analysis software (professional version of Grammarly) for inaccuracies. 

2. Please check the row for age in the patient characteristics table.

The data entered was incorrect; thank you very much for bringing it to our notice. The entire table has been thoroughly reviewed and corrected.(page 6-7)

3. Even though you point to a previous study, I find the paper is lacking some high-level explanation of the methodology (how was the train/test split performed, which feature selection algorithms were employed, what are the weights in the F1-score computation, etc). Right now it is very hard for the reader to understand what methodology was used in this study.

The methodology section has been updated to describe the workflow in a more coherent manner.(page 3-6)

4. I have concerns whether the follow-up time is too little to be clinically significant.

We agree that the follow-up duration, while sufficient for reporting early response, is indeed an inadequate represent of the overall clinical picture. The same has been reiterated in the final paragraph of the discussion stating the limitations of our study.(page 13-14)

Reviewer #2: 

1) How was the annotation of the PT and the LN done? Manually? By whom? was each LN delineated separately?

The annotations of PT and LN were performed using the 3D slicer tool manually by the Radiation Oncologist. The same has been mentioned in the manuscript (page 4).

2)Please provide more details on the extraction of features:

- how many features were extracted per feature type?

We extracted features under six major domains, including shape-based (14), gray-level dependence matrix (14), gray-level cooccurence matrix (24), first-order statistics (18), gray-level run length matrix (16), gray-level size zone (16) and neighboring gray-tone difference matrix (5). The same has been mentioned in the manuscript (page 4-5).

-Which pre-processing was done?

No separate pre-processing of the images was performed; radiomics features were obtained after annotating the CT images with primary and nodal volumes with the pyradiomics toolbox (an extension provided by 3D-slicer annotation tool). Prior to subjecting the data to ML training algorithms, standard scalar was performed on all the variables.(page 5-6)

- Were cubic voxels used?

Yes. Shape-based 3D-radiomics features were used in the analysis. 

- Which binning method?

No binning method used, as the structured radiomics features were directly extracted from the Pyradiomics toolbox.

3) Why did you not split the dataset into a training and validation dataset? Since this is not performed a cross-validation is needed.

The dataset was split into training and testing dataset in the ratio of (70:30). The details are illustrated in the updated methodology section (). Thereafter, cross-validation was performed on complete data for all the models using stratified K-fold cross validation (page 5-6)

4) To better address all these questions please determine the radiomics quality score:

https://www.radiomics.world/rqs and put your score to the manuscript including your answers as an appendix.

Thank you for this suggestion! On attempting the suggested questionnaire, the determined score was, unfortunately, only 45%. However, we would like to state here that at least six of the 16 questions (No.s 3,4,7,8,11 and 15) are not applicable to our study. (Appendix) 

5) Results: With clinico-radiomic data, the mean training and testing accuracy->From Materials and Methods it is not clear that you have splitted in training and test set, please explain.

The dataset was split into training-testing ratio of 70:30, and the means of 10 such iterations were the performance was reported by evaluating the designed model using test dataset. (page 8-12)

6) All your results on accuracy, sensitivity,... need confidence intervals.

Thank you again for pointing this out! Confidence intervals values have been added to all the metrics in the original manuscript. (page 8-12)

---

## [Decision Letter · Decision Letter 1]

6 Oct 2022

PONE-D-22-16089R1Utility of adding Radiomics to clinical features in predicting outcomes of radiotherapy for Head and Neck Cancer using Machine LearningPLOS ONE

Dear Dr. Kadavigere,

Thank you for submitting your manuscript to PLOS ONE. After careful consideration, we feel that it has merit but does not fully meet PLOS ONE’s publication criteria as it currently stands. Therefore, we invite you to submit a revised version of the manuscript that addresses the points raised during the review process.

Dear authors,

thank you for addressing all comments.

Please answer the minor issues raised from reviewer 1 and then the manuscript is suitable for acceptance.

We look forward to receiving your revised manuscript.

Kind regards,

Panagiotis Balermpas

Academic Editor

PLOS ONE

Journal Requirements:

Additional Editor Comments:

Dear authors,

thank you for addressing all comments.

Please answer the minor issues raised from reviewer 1 and then the manuscript is suitable for acceptance.

Reviewers' comments:

Reviewer's Responses to Questions

**Comments to the Author**

1. If the authors have adequately addressed your comments raised in a previous round of review and you feel that this manuscript is now acceptable for publication, you may indicate that here to bypass the “Comments to the Author” section, enter your conflict of interest statement in the “Confidential to Editor” section, and submit your "Accept" recommendation.

Reviewer #1: (No Response)

Reviewer #2: All comments have been addressed

2. Is the manuscript technically sound, and do the data support the conclusions?

Reviewer #1: Partly

Reviewer #2: Yes

3. Has the statistical analysis been performed appropriately and rigorously? 

Reviewer #1: Yes

Reviewer #2: Yes

4. Have the authors made all data underlying the findings in their manuscript fully available?

Reviewer #1: No

Reviewer #2: No

5. Is the manuscript presented in an intelligible fashion and written in standard English?

Reviewer #1: Yes

Reviewer #2: Yes

6. Review Comments to the Author

Reviewer #1: 1. Please check capitalization rules in English and change the capitalized letters accordingly throughout the whole manuscript (e.g., in one sentence you write "artificial intelligence" and "Machine Learning").

2. It is not clear what you mean by "Though the minimum sample size was estimated to be 256, we included all the eligible patients treated between 2013 -2018".

3. "Synthetic samples were taken out from the original samples. The training: testing split was

performed with a 70:30 ratio on the class having majority samples. The training dataset was

generated by adding the train-split of majority samples to the synthetic data, and the testing

dataset was generated by adding test-split of the majority samples to the original minority

samples".

Please rephrase this paragraph and clearly state the class imbalance on the training set and on the test set for each clinical endpoint.

4. Please specify the weights of the F1-score calculation and also provide the non-weighted F1-score.

5. The LRC model is heavily overfitted. Could you implement any measure to prevent this from happening? Such as other feature reduction techniques.

Reviewer #2: Thanks. It is a pitty that the data cannot be made fully available. Please consider for th future to apply for this. I understand this is not easy in medicine but it should still be an aim. I appriciate the calculation of radiomics quality score, It has an average score, the score is important that the reader can easily understand if the manuscript is more hypothesis generating or statisticall significant.

7. PLOS authors have the option to publish the peer review history of their article (what does this mean?). If published, this will include your full peer review and any attached files.

Reviewer #1: No

Reviewer #2: No

---

## [Author Response · Author response to Decision Letter 1]

21 Oct 2022

Date: Oct 06 2022 11:46AM

To: "Rajagopal Kadavigere" rajarad@gmail.com

From: "PLOS ONE" plosone@plos.org

Subject: PLOS ONE Decision: Revision required [PONE-D-22-16089R1]

PONE-D-22-16089R1

Utility of adding Radiomics to clinical features in predicting outcomes of radiotherapy for Head and Neck Cancer using Machine Learning

PLOS ONE

The authors would again like to thank the reviewers profusely for critically reviewing the manuscript and highlighting the points which have helped us considerably improve the quality of the manuscript. The answers to the issues raised by the reviewers have been addressed below. 

The reference list is updated and checked for its correctness. 

Additional Editor Comments:

Dear authors,

thank you for addressing all comments.

Please answer the minor issues raised from reviewer 1 and then the manuscript is suitable for acceptance.

Reviewers' comments:

Reviewer's Responses to Questions

Comments to the Author

1. If the authors have adequately addressed your comments raised in a previous round of review and you feel that this manuscript is now acceptable for publication, you may indicate that here to bypass the “Comments to the Author” section, enter your conflict of interest statement in the “Confidential to Editor” section, and submit your "Accept" recommendation.

Reviewer #1: (No Response)

Reviewer #2: All comments have been addressed

2. Is the manuscript technically sound, and do the data support the conclusions?

Reviewer #1: Partly

Reviewer #2: Yes

3. Has the statistical analysis been performed appropriately and rigorously?

Reviewer #1: Yes

Reviewer #2: Yes

4. Have the authors made all data underlying the findings in their manuscript fully available?

Reviewer #1: No

Reviewer #2: No

5. Is the manuscript presented in an intelligible fashion and written in standard English?

Reviewer #1: Yes

Reviewer #2: Yes

6. Review Comments to the Author

Reviewer #1: 1. Please check capitalization rules in English and change the capitalized letters accordingly throughout the whole manuscript (e.g., in one sentence you write "artificial intelligence" and "Machine Learning").

The sentences were checked thoroughly for capitalization and corrected throughout the manuscript.

2. It is not clear what you mean by "Though the minimum sample size was estimated to be 256, we included all the eligible patients treated between 2013 -2018".

The minimum sample size was calculated for this study as 256 using the proportions of relative hazards formula. The sample size calculation is shown in section 9.2 Appendix. This was the minimum sample size required; however, we screened all the 482 records of patients treated between 2013-18 and selected all the 311 eligible patient records, because a larger sample size is expected to provide better performing models. 

3. "Synthetic samples were taken out from the original samples. The training: testing split was

performed with a 70:30 ratio on the class having majority samples. The training dataset was

generated by adding the train-split of majority samples to the synthetic data, and the testing

dataset was generated by adding test-split of the majority samples to the original minority

samples".

Please rephrase this paragraph and clearly state the class imbalance on the training set and on the test set for each clinical endpoint.

The paragraph has been simplified and rephrased in the manuscript. Also, the number of samples for the training and testing dataset is specified for each clinical endpoints in table 2, 3,4 and 5.

4. Please specify the weights of the F1-score calculation and also provide the non-weighted F1-score.

In the manuscript, Tables 2,3,4 and 5 have been updated with Training and testing f1 score, macro f1 score and weighted f1 score for each class (label 0 and 1). The number of samples for Training and testing class label 0 and 1 serves as the weights to calculate weighted f1 score. The calculation for each performance metrics is presented in section 9.3 appendix. 

5. The LRC model is heavily overfitted. Could you implement any measure to prevent this from happening? Such as other feature reduction techniques.

In this research we have applied intrinsic pre-processing steps to clean the dataset and balance for the class labels. In order to prevent overfitting for locoregional recurrence models, we tried the following additional steps:

1) Computed Principal Component Analysis(PCA) for the original dataset without and with feature selection using Sequential Forward Floating Selection and visualised the results for two principal components. However, a clear class boundary which separates two variables couldn’t be drawn. The PCA plots are shown as follows:

a) b)

Figure: PCA plot of two components for only clinical data a) without feature selection and b) with feature selection

a) b)

Figure: PCA plot of two components for clinico radiomics data a) without feature selection and b) with feature selection

a) b)

Figure: PCA plot of two components for only radiomics data a) without feature selection and b) with feature selection

2) Originally have used the ‘accuracy’ hyperparameter for selecting optimal features. We have redone the analysis for locoregional models by changing the hyperparameter setting to ‘f1’and ‘f1_weighted’

3) Finally, we tried to vary the learning rate ‘C’ in KSVM for Only Clinical and Clinico-Radiomics dataset and ‘max_depth’ in RF for Only Radiomics dataset. The training and testing accuracy variation with the variability of learning rate parameters is shown in the table as follows:

 Hyper Parameter variation Only clinical KSVM Clinico radiomics 

KSVM

KSVM C= 0.1 Training Weighted F1:0.53

Testing Weighted F1: 0.33 Training Weighted F1: 0.44

Testing Weighted F1: 0.25

 C= 0.2 Training Weighted F1:0.69

Testing Weighted F1:0.48 Training Weighted F1: 0.66

Testing Weighted F1: 0.41

 C= 0.3 Training Weighted F1:0.76

Testing Weighted F1: 0.49 Training Weighted F1: 0.86

Testing Weighted F1: 0.53

 C= 0.4 Training Weighted F1: 0.95

Testing Weighted F1: 0.50 Training Weighted F1: 0.99

Testing Weighted F1: 0.53

 C= 0.5 Training Weighted F1: 0.96

Testing Weighted F1: 0.52 Training Weighted F1:0.99

Testing Weighted F1: 0.59

 C= 0.6 Training Weighted F1:0.96

Testing Weighted F1: 0.55 Training Weighted F1: 0.99

Testing Weighted F1: 0.66

 C= 0.7 Training Weighted F1: 0.97

Testing Weighted F1: 0.57 Training Weighted F1: 1

Testing Weighted F1: 0.82

 C= 0.8 Training Weighted F1:0.96

Testing Weighted F1: 0.73 Training Weighted F1:1

Testing Weighted F1:0.81

 C= 0.9 Training Weighted F1:0.97

Testing Weighted F1: 0.72 Training Weighted F1:1

Testing Weighted F1:0.81

 C= 1.0 Training Weighted F1:0.97

Testing Weighted F1: 0.72 Training Weighted F1:1

Testing Weighted F1:0.81

 C= 1000 Training Weighted F1:1

Testing Weighted F1: 0.76 Training Weighted F1:1

Testing Weighted F1:0.78

RF max_depth=5 Training Weighted F1:0.84

Testing Weighted F1: 0.65

 max_depth=10 Training Weighted F1:0.86

Testing Weighted F1: 0.68

 max_depth=50 Training Weighted F1:0.86

Testing Weighted F1: 0.66

 max_depth=100 Training Weighted F1:0.86

Testing Weighted F1: 0.64

 max_depth=1000 Training Weighted F1:0.85

Testing Weighted F1: 0.65

Thus, we concluded that class labels have high overlap due to limitation of low number of samples, the algorithm was bound to overfit. With a greater sample size and higher numbers of positive samples, it might be feasible to prevent this. 

Reviewer #2: Thanks. It is a pity that the data cannot be made fully available. Please consider for th future to apply for this. I understand this is not easy in medicine but it should still be an aim. I appriciate the calculation of radiomics quality score, It has an average score, the score is important that the reader can easily understand if the manuscript is more hypothesis generating or statistically significant.

7. PLOS authors have the option to publish the peer review history of their article (what does this mean?). If published, this will include your full peer review and any attached files.

Do you want your identity to be public for this peer review? For information about this choice, including consent withdrawal, please see our Privacy Policy.

Reviewer #1: No

Reviewer #2: No

---

## [Decision Letter · Decision Letter 2]

24 Oct 2022

Utility of adding Radiomics to clinical features in predicting outcomes of radiotherapy for Head and Neck Cancer using Machine Learning

PONE-D-22-16089R2

Dear Dr. Kadavigere,

We’re pleased to inform you that your manuscript has been judged scientifically suitable for publication and will be formally accepted for publication once it meets all outstanding technical requirements.

Kind regards,

Panagiotis Balermpas

Academic Editor

PLOS ONE

Additional Editor Comments (optional):

Reviewers' comments:

Reviewer's Responses to Questions

**Comments to the Author**

1. If the authors have adequately addressed your comments raised in a previous round of review and you feel that this manuscript is now acceptable for publication, you may indicate that here to bypass the “Comments to the Author” section, enter your conflict of interest statement in the “Confidential to Editor” section, and submit your "Accept" recommendation.

Reviewer #1: All comments have been addressed

2. Is the manuscript technically sound, and do the data support the conclusions?

Reviewer #1: Yes

3. Has the statistical analysis been performed appropriately and rigorously? 

Reviewer #1: Yes

4. Have the authors made all data underlying the findings in their manuscript fully available?

Reviewer #1: No

5. Is the manuscript presented in an intelligible fashion and written in standard English?

Reviewer #1: Yes

6. Review Comments to the Author

Reviewer #1: The authors have addressed my comments and concerns and the manuscript is now adequate and complete to publish.

7. PLOS authors have the option to publish the peer review history of their article (what does this mean?). If published, this will include your full peer review and any attached files.

Reviewer #1: No

---

## [Editor Report · Acceptance letter]

17 Nov 2022

PONE-D-22-16089R2 

Utility of Adding Radiomics to Clinical Features in Predicting the Outcomes of Radiotherapy for Head and Neck Cancer Using Machine Learning 

Dear Dr. Kadavigere:

I'm pleased to inform you that your manuscript has been deemed suitable for publication in PLOS ONE. Congratulations! Your manuscript is now with our production department. 

Kind regards, 

on behalf of

Dr. Panagiotis Balermpas 

Academic Editor

PLOS ONE